# The Influence of the Selection at the Amino Acid Level on Synonymous Codon Usage from the Viewpoint of Alternative Genetic Codes

**DOI:** 10.3390/ijms24021185

**Published:** 2023-01-07

**Authors:** Konrad Pawlak, Paweł Błażej, Dorota Mackiewicz, Paweł Mackiewicz

**Affiliations:** Department of Bioinformatics and Genomics, Faculty of Biotechnology, University of Wroclaw, ul. Joliot-Curie 14a, 50-383 Wroclaw, Poland

**Keywords:** amino acid, codon, codon usage, genetic code, mutations, selection

## Abstract

Synonymous codon usage can be influenced by mutations and/or selection, e.g., for speed of protein translation and correct folding. However, this codon bias can also be affected by a general selection at the amino acid level due to differences in the acceptance of the loss and generation of these codons. To assess the importance of this effect, we constructed a mutation–selection model model, in which we generated almost 90,000 stationary nucleotide distributions produced by mutational processes and applied a selection based on differences in physicochemical properties of amino acids. Under these conditions, we calculated the usage of fourfold degenerated (4FD) codons and compared it with the usage characteristic of the pure mutations. We considered both the standard genetic code (SGC) and alternative genetic codes (AGCs). The analyses showed that a majority of AGCs produced a greater 4FD codon bias than the SGC. The mutations producing more thymine or adenine than guanine and cytosine increased the differences in usage. On the other hand, the mutational pressures generating a lot of cytosine or guanine with a low content of adenine and thymine decreased this bias because the nucleotide content of most 4FD codons stayed in the compositional equilibrium with these pressures. The comparison of the theoretical results with those for real protein coding sequences showed that the influence of selection at the amino acid level on the synonymous codon usage cannot be neglected. The analyses indicate that the effect of amino acid selection cannot be disregarded and that it can interfere with other selection factors influencing codon usage, especially in AT-rich genomes, in which AGCs are usually used.

## 1. Introduction

The redundancy of the standard genetic code (SGC) is a direct consequence of the fact that 64 codons encode 20 amino acids and the signal for the termination of protein synthesis (translation). Therefore, groups of codons exist that encode the same genetic information. In most cases, codons belonging to the given group differ in one position. As a consequence, some point mutations, which may occur between codons in the group, are silent, i.e., they do not change coded information. These mutations and codons are called synonymous. For example, there are codon groups named fourfold degenerated (4FD), in which the third codon position can be changed to any nucleotide without consequences on the encoded amino acid. Such codons encode alanine (Ala), glycine (Gly), proline (Pro), threonine (Thr) and valine (Val) in the SGC. Nevertheless, the distribution of synonymous codons’ usage in protein-coding sequences is not uniform, and some preferences in this usage are observed. This phenomenon is well-known as a codon usage bias [1]. Causes and implications of this bias have been widely investigated and discussed [2,3,4].

There are two main explanations of the codon bias, namely, directional mutational pressure and specific selection [2,3,4]. The mutational view assumes that variability in the usage of synonymous codons results from the mutational processes, which substitute nucleotides with different rates. Consequently, we should expect more codons composed of nucleotides generated more frequently. In the light of this assumption, GC/AT content is the main force responsible for the pattern of codon usage in genomes [5,6,7]. For example, in the genomes characterized by a high GC content, synonymous codons ending with guanine and cytosine are more frequently observed in comparison to those ending with thymine and adenine. It should also be noted that the mutational pressure can be different in individual genome regions [6,8,9] and depend on differently replicating DNA strands, i.e., the leading and the lagging strands [10,11,12,13,14]. Thus, genes located in various genomic regions and DNA strands can demonstrate a specific codon usage.

The second opinion points out that the synonymous codon bias arises from selection factors. For example, it was shown that highly expressed protein-coding genes have a tendency to use codons for which there are more tRNA molecules in a cell [15,16,17,18,19,20,21,22]. This codon bias is supposed to be an adaptation to a more effective, faster and accurate translation process.

The variable usage of synonymous codons was also detected in different regions of protein-coding sequences. It was explained by selection against the formation of mRNA secondary structures at the 5${}^{\prime }$ end to ensure the efficient initiation of translation [23] or by selection for the optimal protein folding and adopting the correct structure during the translation [24,25,26]. What is more, some poorly adapted codons in the first 90–150 codon positions in genes are expected to slow down the translation process to reduce the number of ribosomal blockages and collision-induced abortions [27]. It was also proposed that the codon usage changes the gene expression by the influence on the transcription process [28,29,30] and mRNA stability [31].

Morton [32] proposed another important factor influencing the synonymous codon usage in protein-coding genes. He postulated that this effect results from a general selection at the amino acid level and associated with that a biased selection for codon substitutions. For example, glycine is encoded by four codons, namely, GGA, GGC, GGT and GGG in the standard genetic code. Its codon GGA can potentially mutate by a single substitution into TGA, which encodes a translation termination signal causing premature protein synthesis. Other glycine codons have no risk to be changed into the stop codon by single-nucleotide mutations. Since the GGA→TGA mutation is deleterious, it is not accepted by selection. Thus, the codon GGA would not disappear, at least due to these mutations, as would other Gly codons whose changes to other codons are more tolerated. In consequence, the frequency of GGA should be higher than the other codons. If the codon TGA encodes Gly or other similar amino acid, we should expect a greater tolerance for the GGA→TGA mutation and a similar disappearance rate of GGA as other glycine codons. Then, the usage of all GGN codons would be more balanced. Actually, there are the alternative genetic codes in which TGA codes for an amino acid and this effect should be observed. Of course, the frequency of these codons is the result of many other substitutions, which are not equivalent from a selectional point of view. Thus, the diverse selection against substitutions can lead to distinct balances between the loss and generation of the codons and can finally cause their different usage.

In fact, Morton [32] showed that the composition of the third positions in synonymous codons deviates from the composition of non-coding neutral sites, even in the absence of direct selection on the synonymous codons. However, his study only considered four selected mutational processes generating equal frequencies of complementary nucleotides. Therefore, this hypothesis was further explored by Błażej et. al. [33], who developed a theoretical mutation–selection model model and examined 88,560 nucleotide stationary distributions produced by mutational pressures under selection at the amino acid level. They found that the influence of selection at the amino acid level is visible, especially for theoretical mutational pressures generating more adenine and thymine than guanine and cytosine, as well as more purines than pyrimidines. Therefore, this effect cannot be neglected and requires further studies.

The previous study [33] analyzed the synonymous codon usage assuming the assignment of amino acids to codons in the standard genetic code (SGC), which is nearly universal among all living organisms. However, there are deviations from this universality, called alternative genetic codes (AGCs). These genetic coding systems operate mainly in small mitochondrial [34,35,36,37,38,39] and plastid genomes [40,41,42], but also in the genomes of parasites and symbionts [43,44,45,46] or even the nuclear genomes of various eukaryotes [47,48,49,50,51,52,53].

Generally, the deviations from the SGC observed in the AGCs can result from: (i) the reassignment of codons encoding the typical 20 amino acids and stopping the protein translation; (ii) loss of codon meaning due to the disappearance of this very codon; and (iii) the incorporation of new amino acids, e.g., selenocysteine and pyrrolysine [54,55,56]. The deviations of the first type are the most frequent. The changes in the codon’s meaning are mainly related to reassignments of the stop to sense codons, e.g., TGA to tryptophan. In the study, we analyzed all AGCs available at the NCBI database that were characterized by different assignments of amino acids and/or stop translation signals to codons than in the SGC. They also had the four amino acids, Ala, Gly, Pro, Thr and Val, coded by the same 4FD codons as the SGC.

Because the acceptance of a given codon substitution to others depends on the assignment of amino acids to codons, we can expect that the alternative genetic codes can demonstrate a different influence on codon usage than the SGC. Therefore, we decided to study the effect of selection at the amino acid level on codon usage, including the alternative codes. The comparison of AGCs with the SGC revealed that the alternatives minimize consequences of point mutations or translational errors much better [56,57], and the SGC is not fully optimized in this respect [58,59,60,61,62,63,64,65,66]. Thus, we can also expect a different strength of selection at the amino acids level on the codon usage in these alternative coding systems.

## 2. Results

The main goal of our investigation was to assess a potential selection strength at the amino acid level on the usage of codons, assuming coding systems in alternative genetic codes and in the standard genetic code for comparison. Therefore, we calculated the usage of fourfold degenerated (4FD) codons generated under almost 90,000 nucleotide stationary distributions generated by mutational processes and compared it with the usage of codons whose mutations to other codons were subjected to selection based on differences in the physicochemical properties of coded amino acids. The differences were described by Grantham’s chemical similarity matrix [67].

To estimate the selection strength on the codon usage $F_{\pi }$, we calculated for each of 4FD codons the difference between the codon frequency after the selection and their expected frequency resulting only from mutations. The difference was divided by the expected frequency. Next, we summed up the differences obtained for five 4FD codons. The high $F_{\pi }$ values should be interpreted as a large variation in the synonymous codon usage due to selection at the amino acid level. Because a stationary nucleotide distribution can be realized by many mutational pressures, we applied a modified evolutionary optimization algorithm to determine the pressures for which the effect on the relative codon usage was maximized.

### 2.1. General Performance of Genetic Codes in Terms of the Selection Strength at the Amino Acid Level on the Codon Usage

Based on conducted simulations for each genetic code, we obtained 88,560 values of selection strength on the codon usage $F_{\pi }$ produced under various mutational pressures generating various stationary nucleotide compositions. In order to present a general performance of the codes in terms of the selection strength on codon usage, we calculated several descriptive statistics of $F_{\pi }$ such as the maximum, the minimum and the median, as well as the percentage of changes in relation to the SGC (Table 1, Figure 1).

As one can notice, the selection strength is different for the individual genetic codes. On one hand, there are ten AGCs that show a lower median $me(F_{\pi })$ among all tested mutational stationary distributions than the SGC. Three of them, i.e., codes 9, 14 and 21, have a 19–21% smaller median than the SGC (Table 1, Figure 1). These codes are used in the mitochondria of flatworms (codes 9 and 14), roundworms (code 14), trematodes (code 21) and echinoderms (code 9). The results suggest that the synonymous codon bias generated under these codes due to the selection at the amino acid level is generally smaller than in the SGC. These codes produce also the maximum selection strength on the codon usage $max(F_{\pi })$, which is 25%-31% lower than the SGC. Thereby, they induce a weaker selection on the codon usage under the studied mutational pressures.

On the other hand, the rest of the 13 investigated alternative genetic codes are characterized by a larger $me(F_{\pi })$ in comparison to the SGC. Five of them, i.e., 22, 23, 2, 12 and 26, can generate a codon bias 12% to 37% larger than the SGC (Table 1, Figure 1). Three codes were found to operate in the mitochondrial genomes of various eukaryotes: code 22 in green alga *Scenedesmus obliquus*, code 23 in a protist labyrinthulid *Thraustochytrium* and code 2 in vertebrates, whereas code 12 and 26 is an alternative to the yeast nuclear genome. Their values of $min(F_{\pi })$ and $max(F_{\pi })$ are also substantially higher, from 6% to 75%, than in the SGC. This indicates that the bias in the synonymous codon usage induced by these codes under the selection at the amino acid level is generally greater than in the SGC.

### 2.2. The Importance of Individual Assignments in Alternative Genetic Codes in the Selection Strength on Codon Usage

It is interesting to check which assignments of amino acids to codons in AGCs contribute the most to decreasing or increasing the selection strength on the codon usage. Codes 9, 14 and 21, characterized by a lower median $me(F_{\pi })$ than the SGC, feature the reassignment of stop codon TGA into tryptophan. In code 14, the stop codon TAA also encodes tyrosine. The codes include the reprogramming of several other codons into canonical amino acids, too.

It can be noticed that the codon TGA is redefined to Trp in these codes, which results in the single-nucleotide substitution of GGA (Gly) to TGA being more acceptable from the selection point of view than in the SGC, in which TGA encodes a stop translation signal. This reassignment can decrease differences in the loss and generation of glycine codons and, finally, in their relative usage (Figure 2). Three codes, 9, 14 and 21, are also characterized by the reassignment of arginine codons AGA and AGG into serine, which results in all four codons in the group AGN encode Ser. In consequence, the single nucleotide mutations of individual 4FD codons from the group Thr (ACN) and Gly (GGN) to these serine codons have a similar selective value, which can equalize their relative frequency (Figure 2). Of course, other codon redefinitions can also indirectly modify the usage of the fourfold degenerated codons considered here. Consequently, the individual selection strength on the usage of Thr and Gly codons $F_{\pi |s}$ (see Material and Methods) is smaller for codes 9, 14 and 21 than for the SGC. In the case of ACN codons this measure is 0.049, 0.047 and 0.037 for codes 9, 14 and 21, whereas 0.053 for the SGC. For GGN codons, $F_{\pi |s}$ is 0.041, 0.046 and 0.021 for the AGCs and 0.049 for the SGC.

On the other hand, the alternative coding systems 22, 23, 2, 12 and 26 are characterized by a substantially higher $me(F_{\pi })$ of selection strength on the 4FD codon usage. In code 22, the codon TCA means the stop translation signal instead of serine as other codons in the TCN group. This codon redefinition entails unbalanced substitutions of alanine codons GCN differing from the serine codons TCN by the first codon position (Figure 3). The deleterious mutation of GCA (Ala) to TCA (stop) can be accepted by selection with much lower probability in the comparison to substitutions of other Ala codons. Consequently, the Ala codons can be unequally used. In fact, $F_{\pi |s}$ for GCN is much higher for code 22 than for the SGC, i.e., 2.677 vs. 0.706.

Likewise, another Ser codon group AGN is disrupted in code 2 because its codons AGA and AGG encode the stop signal. In this case, the acceptance of substitutions of threonine codons ACA and ACG into these new stop codons can be weaker than substitutions of other Thr codons ACT and ACC into intact Ser codons AGT and AGC, which influences the unbalanced frequency of Thr codons (Figure 3). Therefore, $F_{\pi |s}$ for ACN is 3.953 in the case of code 2 and only 0.670 for the SGC.

In turn, code 23 has the TTA codon encoding the stop translation signal instead of leucine. Therefore, the substitution of the valine codon GTA into TTA can be much less frequently accepted than substitutions of other Val codons: GTG into TTG still encoding leucine, as well as GTT and GTC, into TTT and TTC, respectively, coding for phenylalanine (Figure 3). All three amino acids, Val, Leu and Phe, show similar hydrophobic properties, so their substitutions are much more easily acceptable. In consequence, the equality in the frequency of Val codons can be disturbed, so $F_{\pi |s}$ for Val codons for code 23 is 3.264 and larger than 1.369 for the SGC.

A similar explanation can be applied to proline codons (CCN) in code 12, in which CTG encodes Ser rather than Leu similar to other codons in the CTN group. The substitution CCG (Pro)→CTG (Ser) can be less accepted than substitutions of Pro to Leu: CCA→CTA, CCC→CTC and CCT→CTT, because, according to general physicochemical properties, proline is more similar to leucine than serine (Figure 3). In agreement with that, $F_{\pi |s}$ for Pro codons is 2.557 for code 12 and is greater than 0.473 for the SGC.

Finally, in code 26, CTG encodes Ala instead of Leu as other CTN codons. These amino acids show distinct physicochemical properties. For example, alanine is much smaller than leucine. Thus, the mutations of proline codons CCN and valine codons GTN into CTN codons in this alternative code have different selection values. CCT→CTT and GTT→CTT can be less accepted than substitutions of other Pro or Val codons into unchanged Leu codons (Figure 3). Thus, $F_{\pi |s}$ values for the CCN (2.641) and GTN (2.641) codons are greater in code 26 than in the SGC (0.473 and 1.369, respectively).

### 2.3. The Influence of Mutational Pressure on the Selection Strength on Codon Usage

The selection strength at the amino acid level $F_{\pi }$ acting on the 4FD codon usage should depend on the rate of individual nucleotide substitutions, which can change the general codon frequency. In Figure 4, we compared the rate of individual nucleotide substitutions in the mutational pressures that minimize and maximize $F_{\pi }$. The rates of substitutions T→A, T→C, G→A, G→C and C→T were statistically higher (with *p* < 0.05 corrected by the Benjamini–Hochberg procedure and the paired Wilcoxon test) in most mutational pressures responsible for the increase in $F_{\pi }$. This means that these substitutions can elevate the selection strength on the 4FD codon usage. On the other hand, the probability of substitutions A→G, G→T and C→G was statistically higher (with *p* < 0.05) in the mutational pressures that decreased $F_{\pi }$ and can reduce the codon bias.

For example, the substitution G→A can increase the selection strength on the codon usage because it is responsible in most genetic codes for many mutations of 4FD codons into those encoding different amino acids. Because of this substitution, Val codons (GTN) can be mutated into ATN codons encoding Ile or Met in the case of all studied genetic codes. Gly codons (GGN) can mutate due to this substitution into two codon groups encoding various amino acids depending on the genetic code: GAN codons for Asp/Glu (in all studied codes) or AGN codons for Arg/Ser (in codes 1, 3, 4, 6, 10–12, 15, 16, 22, 23, 25–32), Ser/Stop (in code 2), Gly/Ser (in code 13) or Lys/Ser (in codes 24 and 33).

We also investigated the stationary nucleotide distributions generated by the mutational pressures that minimize and maximize the selection strength at the amino acid level $F_{\pi }$. The results are presented in Table 2 and Table 3. Interestingly, the stationary distributions that minimize the selection strength are characterized by a high G+C nucleotide content, from 0.80 to 0.9 (Table 2). For example, the nucleotide stationary distributions associated with code 21 showing the lowest $min(F_{\pi })$ is characterized by the 0.89G+C fraction.

In contrast to that, the stationary distributions that maximize the amino acid selection strength on the 4FD codon usage under the respective genetic codes generate a high A+T content (Table 3). The nucleotide distributions are characterized by the fraction of adenine and thymine from 0.88 to 0.89. The distribution associated with code 23 showing the highest $max(F_{\pi })$ is characterized by the extreme fraction of adenine 0.84, whereas other nucleotides constitute only 0.05 or 0.06.

The relationships of $min(F_{\pi })$ and $max(F_{\pi })$ from the stationary nucleotide distributions are presented in Figure 5. It can be seen that the minimum selection strength on the 4FD codon usage can be obtained by all genetic codes when they are subjected to the mutational pressure generating a very low amount of adenine and thymine. Considering the frequency of guanine and cytosine, we can find two groups of codes. One of these groups (codes 2, 5, 12, 13, 21 and 31) produced a $min(F_{\pi })$ under the stationary distribution that was richer in guanine (0.66–0.83) than cytosine (0.07–0.21), and the other group under the excess of cytosine (0.44–0.75) over guanine (0.15–0.41).

Considering the maximum selection strength on the 4FD codon usage, we can notice that all codes are grouped in the plot for a very low frequency of cytosine and guanine (Figure 5). However, in terms of adenine and thymine content, they are separated into two groups. Most of the codes generated $max(F_{\pi })$ under the pressures producing a high frequency of thymine (0.67–0.82) and a lower frequency of adenine (0.07–0.22). In turn, other codes (1, 2, 16, 22, 23, 25 and 30) subjected to the mutations generating a high content of adenine (0.78–0.84) and a small content of thymine (0.05–0.11) resulted in the maximum $\left(F_{\pi }\right)$. The results indicate that extreme codon usage can be generated under the selection on the amino acid level by mutational pressures with stationary nucleotide distributions specific for various groups of genetic codes.

Interestingly, the stationary distributions that are associated with the minimization of the selection influence on the usage of 4FD codons are more similar to the nucleotide composition of most these codons, i.e., Ala, Gly and Pro, because they are also rich in G+C content (0.83). The similarity is visualized in Figure 6, which presents the biplot of correspondence analysis for the nucleotide composition of 4FD codons and nucleotide stationary compositions generated by mutational pressures, which, together with the amino acid selection, minimize or maximize $F_{\pi }$ for individual genetic codes. The 4FD codon groups for Pro, Ala and Gly are clustered with the compositions associated with the minimum deviation in the relative codon usage. These nucleotide contents are characterized by the high frequency of G and C. The separation of nucleotide compositions associated with the maximization of $F_{\pi }$ is also clearly visible. One set is characterized by the high content of A and the other of T. The codon groups for threonine and valine are located in the middle of the plot due to the equal content of A+T and G+C.

The comparison in Figure 6 indicates that the composition of most 4FD codons remains closer to the equilibrium state with the pressures minimizing the selection strength on the codon usage. In contrast to that, the pressures generating more A+T cause these codons to be more frequently substituted by the codons rich in adenine and thymine. This would explain why the 4FD codon usage is more biased under the mutational pressures generating a high frequency of A and T.

The results showed only the extreme nucleotide distributions that were associated with the minimum and maximum deviation in the 4FD codon usage. Therefore, we investigated the relationships between the maximum deviation in the selection strength, i.e., $max(F_{\pi })$, and the stationary frequency of nucleotides induced by the mutational pressures that, together with the amino acid, generated this highest codon bias selection. The plot in Figure 7 demonstrates that the dependence of median $F_{\pi }$ on the A+T content is monotonically increasing, although non-linearly, for all studied genetic codes. Initially, the $me(F_{\pi })$ increases with the rise of A+T in all considered coding systems to about 0.25 of A+T. It stabilizes to ca 0.60 of A+T and then rapidly grows up to the highest values for the extreme A+T content. The alternative genetic codes 9, 14 and 21 show the lowest course of the curve, whereas the curve of code 22 stands out from the others and achieves the highest position in the plot. This indicates that the most biased 4FD codon usage is produced by this code under mutational pressures, generating a very wide range of A+T compositions.

### 2.4. The Comparison of Calculated Deviations in the Codon Usage with That Observed in Protein-Coding Genes

The results presented above refer to the maximum deviation in the codon usage that can be exerted by theoretical mutational pressures with the selection at the amino acid level for the given genetic codes. Thus, it is interesting to assess the strength of this effect in the case of real biological data. Therefore, we calculated an analogous measure to $F_{\pi }$, called *F*, which is the normalized difference between: (i) the observed usage of 4FD codons in protein-coding sequences that are translated by the studied genetic codes and (ii) the expected frequencies of these codons. The expected frequencies were approximated by the average of relative frequencies of all 4FD codons in these protein-coding genes (See Material and Methods for details). The obtained values were compared with the $F_{\pi }$ values collected from the theoretical calculations (Figure 8).

Although the medians of *F* for biological data are larger than those achieved for the theoretical effect (*p* < 1 × 10${}^{-16}$ corrected by Benjamini–Hochberg procedure, Kruskal–Wallis test), the values are of the same order of magnitude and the distributions overlap (Figure 8). Moreover, some theoretical cases are even greater than those for protein-coding genes. Comparing the median values, we can ascertain that the the selection strength on the 4FD codon usage estimated in the computer simulations constitute about 35% of the deviation in the codon usage found for protein coding sequences read under the studied genetic codes. The deviation in the 4FD codon usage assessed for the protein genes translated by the SGC and the AGCs are comparable and not statistically different from each other (*p* = 0.84).

The bias in the codon usage obtained for the real gene sequences can be higher than those for the theoretical calculations because additional factors, e.g., those associated with the effectiveness of translation, can influence the codon usage. Nevertheless, the results indicate that the effect of selection at the amino acid level can explain a substantial proportion of the observed 4FD codon usage.

## 3. Discussion

The results enabled us to assess the impact of mutational pressures and the selection at the amino acid level on the usage of fourfold degenerated codons. The influence was studied for alternative genetic codes in comparison with the standard genetic code. The analyses showed that there are ten AGCs that generate 4FD codon biases smaller than the SGC. However, 13 AGCs produced a much higher diversity in the 4FD codon usage than the SGC. The codes generating the largest codon bias evolved independently in various eukaryotic groups, in mitochondrial genomes of green algae, labyrinthulids and vertebrates, as well as the nuclear genomes of yeasts. New codes are still being discovered, especially in poorly studied groups of protists [52,68,69,70]. Therefore, we cannot exclude that other codes can show a more extreme influence on the codon usage.

The weaker bias in the 4FD codon usage can be associated with specific codon assignments in AGCs that minimize differences in the loss and generation of codons in a given 4FD group. In turn, the greater dissimilarity in the usage can be associated with a larger disequilibrium between the loss and generation of these codons. This effect should be enhanced when codons in a 4FD group mutate into amino acids that differ substantially from each other in stereochemistry and physicochemical properties. A much greater distinction between the codon usage should be observed when a codon out of this group is substituted into the stop translation signal, whereas other codons in this group are mutated into codons encoding the same amino acid or showing similar properties. In these cases, the difference between probability of accepted substitutions is the largest. Actually, such assignments are present in many AGCs.

The influence of selection at the amino acid level on the 4FD codon usage is clearly related to the mutational pressure acting on the codons and generating a specific nucleotide stationary distribution. The pressures that produce the high frequency of guanine or cytosine decrease the 4FD codon bias. Most 4FD codons are also GC-rich, which can indicate that the mutational pressures generating more G and C hardly change the usage of these codons under the applied selection model. In turn, the pressures generating more thymine or adenine at the expense of guanine and cytosine contribute to the largest difference in the usage of 4FD codons. Thus, these pressures can change the usage of these codons to a greater extent due to a higher rate of some substitutions, e.g., G→A and C→T. It should be emphasized that the diversified usage of 4FD codons is produced after the influence of the selection pressure associated with differences in physicochemical properties of the coded amino acids but not only the pure mutational pressure. The two processes, the mutations and the diverse selection of the codon substitutions, can together generate the high deviation in the usage of 4FD codons. Bearing that in mind, the calculated deviation in codon usage $F_{\pi }$ was normalized by the codon usage generated only by the mutations.

The comparison of 4FD codon usage bias in the real gene sequences with the theoretical estimations demonstrates that the effect of selection at the amino acid level could explain the observed codon usage bias and should not be disregarded. This effect can constitute even more than one-third of the total 4FD codon usage. Our results also have interesting evolutionary consequences and imply that combinations of various mutational pressures and alternative genetic codes can increase and decrease the 4FD codons’ usage. The effect of selection at the amino acid usage is most visible under the mutations generating the high A+T content. Therefore, this influence should be enhanced in AT-rich genomes, which are present in bacterial and eukaryotic endosymbionts and intracellular parasites [71,72,73,74,75,76,77] and most organelles [78,79]. It was also found that even in free-living bacteria, there is a tendency to generate more adenine and thymine [73,80]. This means that the deviation in 4FD codon usage due to the amino acid selection can be more common than one can assume.

The selection at the amino acid level on codon usage can disturb and interfere with the effects of other factors influencing codon usage, e.g., selection on translational speed and efficiency. This implies that cellular systems involved in protein translation should be modified and adapt with the change in mutational pressure and reassignments in the genetic code. The greater deviation of 4FD codon usage in AGCs than the SGC suggests that the translation systems under the alternative genetic systems needs to undergo a major reorganization, if the accuracy and effectiveness of protein synthesis and folding are maintained at the same level.

Codon usage is important for genomes of viruses, which utilize the translational machinery of their host and have to adapt the usage in order to optimize the synthesis of their proteins. The higher similarity of viral codon usage to the host can facilitate the replication of these pathogenic agents. The usage is similar to specific host genes [81,82] and is visible especially in viruses infecting a narrow spectrum of hosts [83,84]. Therefore, dynamic changes in codon usage in the hosts, e.g., due to changes in mutational pressure, can protect them against infectious agents. Then, the speed and efficiency of viral protein synthesis can be lower and worse. Thereby, the hosts can avoid or diminish the infection.

The findings presenting here indicate that the effect of amino acid selection on the codon bias can occur under various coding systems and cannot be neglected and should be elevated under the mutational pressure generating more AT than GC. Disregarding this effect can mislead that only other factors influence the synonymous codon usage. For example, one can overestimate the number of protein-coding genes whose codon usage is subjected to selection on the translational efficiency. It is not inconceivable that the selection at the amino acid level is a primary cause of synonymous codon usage, and other selectional forces have to fit to it. For example, the concentration of tRNA isoacceptors should be adapted to the number of recognized codons in protein-coding sequences, which can be modified by the selection at the amino acid level. When many tRNA isoacceptors recognizing the most frequent synonymous codons are produced, the translational elongation is more effective [19,85].

## 4. Materials and Methods

We included in the study all 23 AGCs that encode 20 canonical amino acids and differ from the SGC in at least one assignment of amino acids and/or stop to codons (Table 4). At the same time, these codes contain the same 4FD codons for Ala, Gly, Pro, Thr and Val as the SGC. The codes were compiled by Andrzej (Anjay) Elzanowski and Jim Ostell at the National Center for Biotechnology Information (NCBI), Bethesda, MD, USA. Detailed information about these coding systems are available at the NCBI database (https://www.ncbi.nlm.nih.gov/Taxonomy/Utils/wprintgc.cgi), accessed on 10 November 2022.

The applied methodology was based on that from [33]. Briefly, to detect the influence of selection at the amino acid level on the usage of 4FD codons for Ala, Gly, Pro, Thr and Val, we compared the usage produced by various mutational pressures with that subjected also to selectional constraints.

The effect of mutations was induced by homogeneous, stationary and continuous-time Markov processes characterized by 88,560 different stationary distributions of four nucleotides A, C, G and T. The stationary distributions of these mutational pressures, i.e., the frequencies of particular nucleotide, ranged from 0.05 to 0.85 with 0.01 increments. For each distribution, we generated rate matrices under an unrestricted (UNREST) model of nucleotide substitutions [86]. After uniformization of the rate matrices to transition probability matrices of nucleotides, we calculated mutational transition probability matrices for codons.

The selection pressure was described by an amino acid acceptance matrix based on Grantham’s distance matrix depending on three physicochemical properties of amino acids: composition, polarity and molecular volume [67]. The values in this matrix were converted into acceptance probabilities. In the case of substitutions involving stop translation signal, we assumed the lowest possible probability of acceptance in the derived acceptance matrix. Including the mutational and selectional components in one matrix, we also calculated mutational–selectional transition probability matrices for codons.

However, it is well-known that there are infinitely many Markov processes for a given stationary distribution. Therefore, we applied a modified version of the Evolutionary Strategies approach [87] in order to detect the maximum differences between the relative 4FD codon usage induced by a given Markov process with and without the effect of selection. Thus, for each mutational process acting together with selection, we calculated the selection strength $F_{\pi }$ in two steps. Firstly, we calculated the normalized difference between the relative frequency of 4FD codons after the selection and their expected frequency resulting only from a mutation process:$$F_{\pi |s}=\sum_{i\in A,T,G,C}\frac{|\pi_{i}-\frac{\pi_{s_{i}}^{sel}}{\pi_{s}^{sel}}|}{\pi_{i}}\;,$$
where $\pi_{i}$ is a stationary frequency of *i* nucleotide; *s* denotes a group of 4FD codons assigned to a specific amino acid; $s_{i}$ is a selected codon belonging to *s* group, where *i* nucleotide occurs at the third codon position. Thus, the measure $F_{\pi |s}$ describes the selection strength in the case of single codon group *s*. Based on that, the selection strength for all 4FD codon groups was obtained:$$F_{\pi }=\sum_{s\in S}F_{\pi |s},$$
where *S* is the set of all considered codon groups.

Moreover, we calculated a parameter corresponding to $F_{\pi }$ for the 4FD codons in protein-coding sequences translated by appropriate genetic codes [33]. This measure *F* is the summarized deviation from the expectation in the codon usage for all 4FD groups in these sequences:$$F=\sum_{s\in S}f_{s},$$
where
$$f_{s}=\sum_{i\in A,T,G,C}\frac{|e_{i}-\frac{o_{s_{i}}}{o_{s}}|}{e_{i}}\;$$
is the normalized difference between its relative frequency of 4FD codons in a group and their expected frequency; $o_{s_{i}}$ is the observed frequency of the 4FD codon with a nucleotide *i* at the third codon position; $o_{s}$ is the observed frequency of all codons in the group; and $e_{i}$ is the expected frequency of the codon calculated as the average of relative frequencies of all 4FD codons with a nucleotide *i* at the third codon position.

The deviations *F* from the expectation in the 4FD codon usage for protein genes translated by the SGC in 4879 bacteria were taken from [33], whereas the deviations *F* for protein genes translated by AGCs were calculated based on the codon usage published by [68] for codes 27 and 28 or derived from sequences of these genes collected from the NCBI GenBank database (https://www.ncbi.nlm.nih.gov/), accessed on 16 April 2022. in the case of other codes. In the analysis, we considered only these codes for which we were able to collect at least 12 protein-coding sequences. Code 30 was excluded from this analysis because we found only two sequences for it. In total, we gathered 637,066 sequences for the studied AGCs.

## Figures and Tables

**Figure 1 ijms-24-01185-f001:**
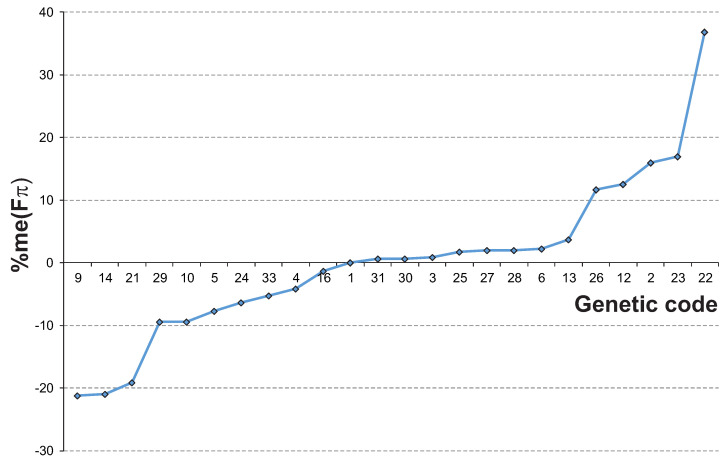
The ranking of alternative genetic codes according to percent changes in the median selection strength on the 4FD codon usage $F_{\pi }$ in relation to the SGC. The genetic codes were numbered according to the NCBI database (https://www.ncbi.nlm.nih.gov/Taxonomy/Utils/wprintgc.cgi), accessed on 10 November 2022. See Materials and Methods for details.

**Figure 2 ijms-24-01185-f002:**
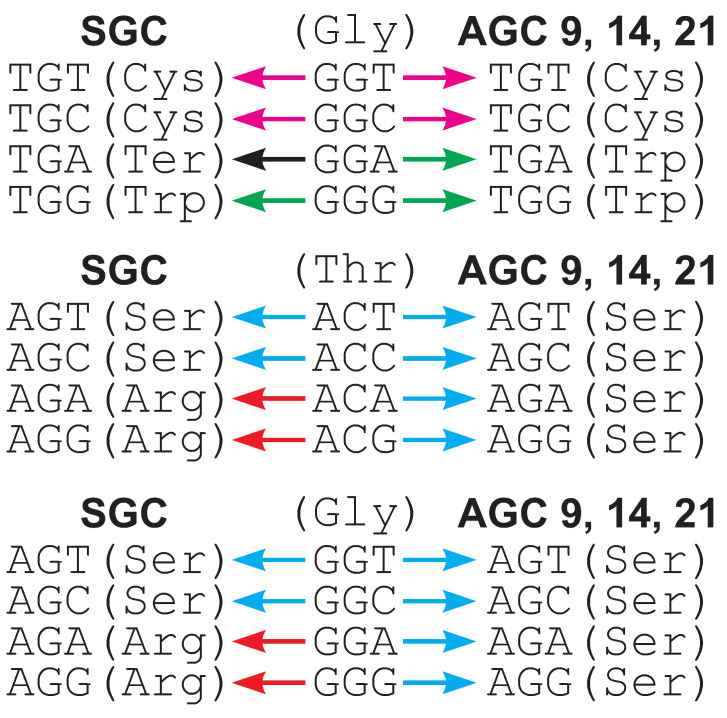
Examples of codon substitutions that can decrease the usage of fourfold degenerated codons in alternative genetic codes (AGCs) in comparison to the standard genetic code (SGC). Colors of arrows correspond to substitutions to a specific amino acid or stop translation signal. In contrast to the SGC, substitutions under AGCs generate codons encoding the same or more similar amino acids.

**Figure 3 ijms-24-01185-f003:**
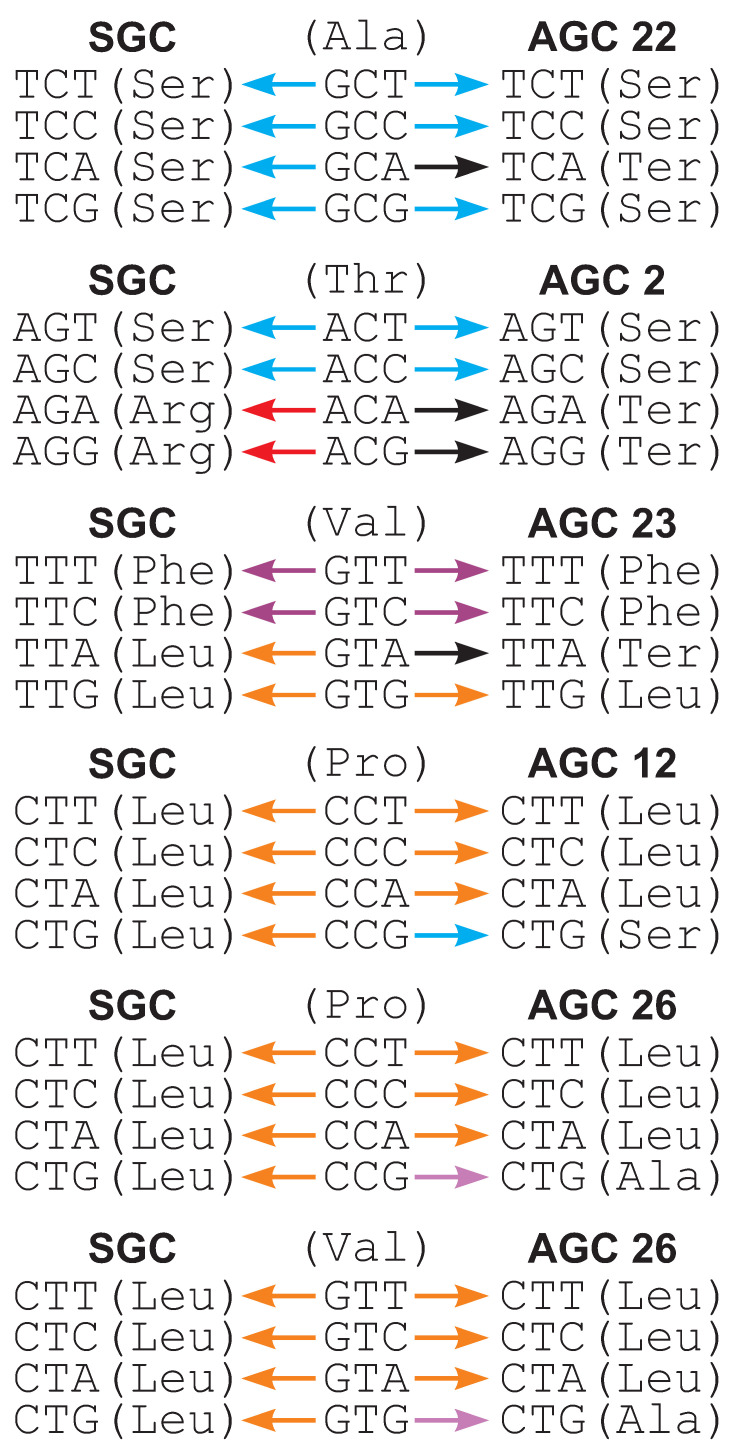
Examples of codon substitutions that can increase the usage of fourfold degenerated codons in alternative genetic codes (AGCs) in comparison to the standard genetic code (SGC). Colors of arrows correspond to substitutions to a specific amino acid or stop translation signal. In contrast to the SGC, substitutions under AGCs generate codons encoding more different amino acids or stop translation signal.

**Figure 4 ijms-24-01185-f004:**
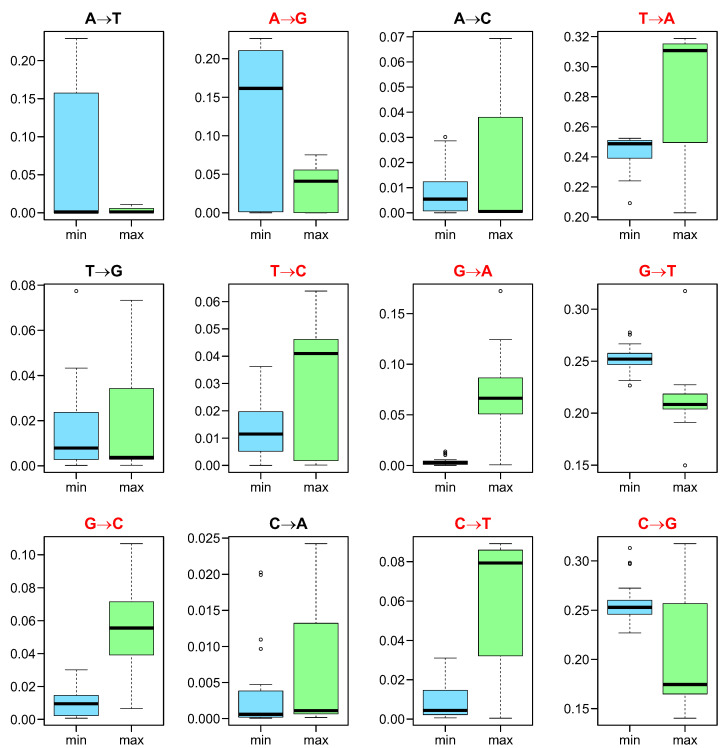
The rates of nucleotide substitutions in the mutational pressures (*Y* axes) which, together with the selection at the amino acid level, minimize (min) or maximize (max) the selection strength $F_{\pi }$ on the 4FD codon usage calculated for known genetic codes. The thick line indicates median, the box shows quartile range and the whiskers denote the range without outliers. Substitutions statistically different with *p* < 0.05 between the compared groups are indicated in red.

**Figure 5 ijms-24-01185-f005:**
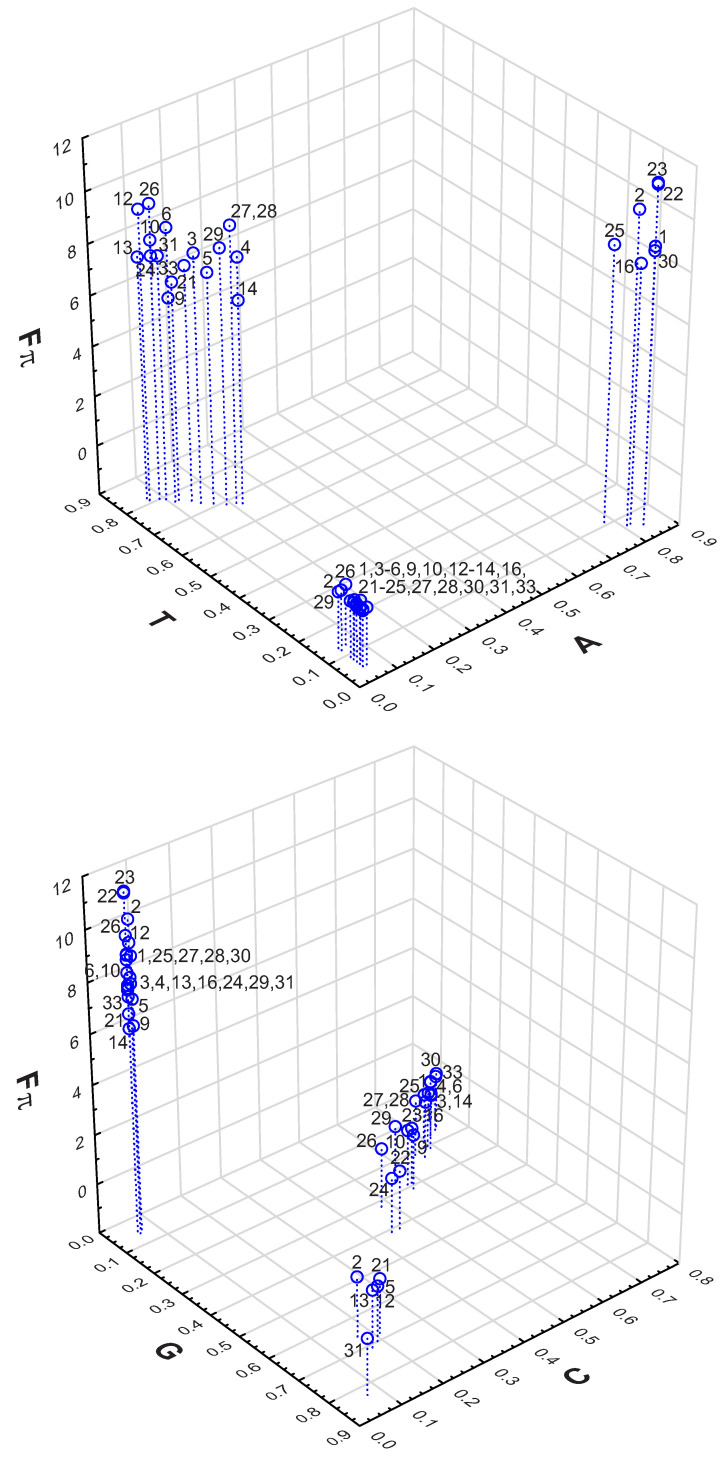
The relationship between the selection strength on codon usage $F_{\pi }$ and the stationary nucleotide composition of adenine (A) and thymine (T), as well as guanine (G) and cytosine (C), produced by mutational pressures, which, together with the selection at the amino acid level, generate the minimum and maximum values of $F_{\pi }$. Labels at points in the plots indicate individual genetic codes for which these values were calculated.

**Figure 6 ijms-24-01185-f006:**
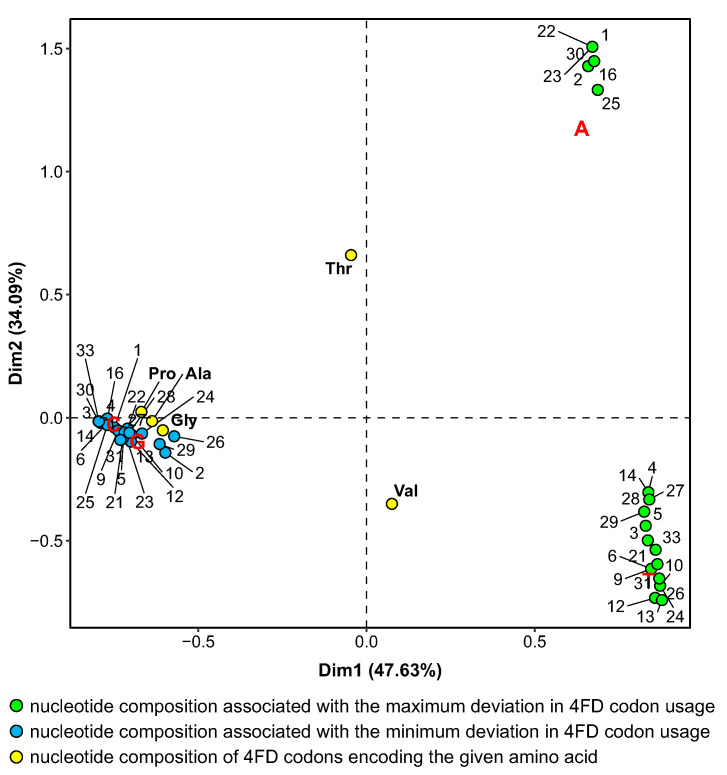
The result of correspondence analysis for stationary nucleotide compositions generated by mutational pressures which, together with the selection at the amino acid level, generate the extreme variation in the 4FD codon usage. Numbers indicate individual genetic codes under which the usage was calculated. Ala, Gly, Pro, Thr and Val are names of amino acids coded by the 4FD codon groups.

**Figure 7 ijms-24-01185-f007:**
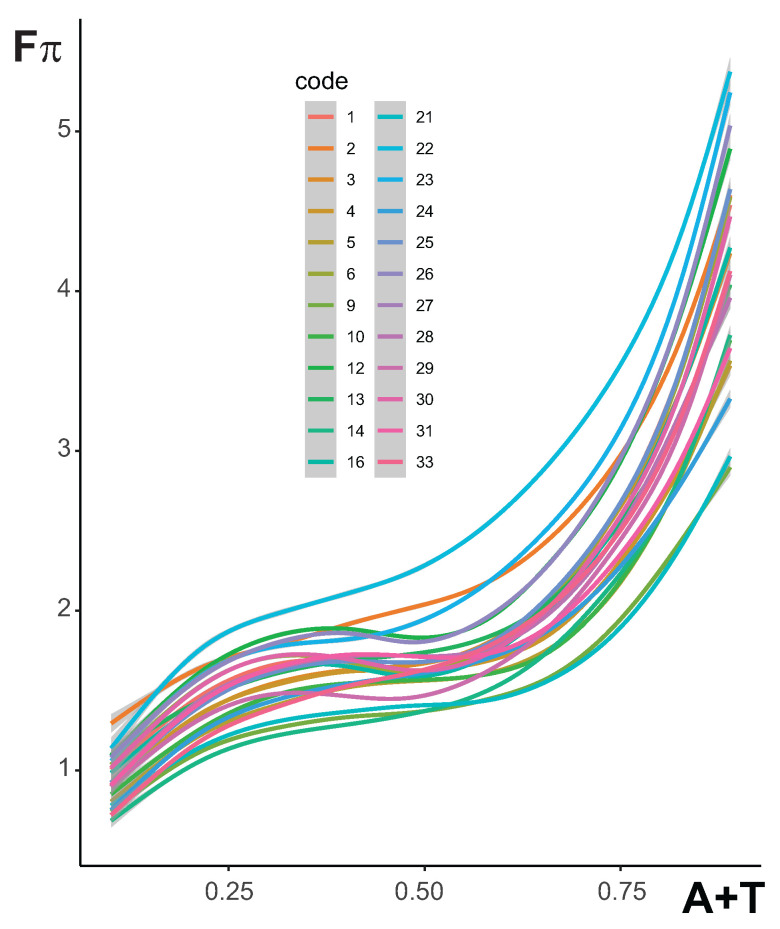
The relationship between the median of $\left(F_{\pi }\right)$ values and the sum of stationary frequencies of adenine (A) and thymine (T) calculated for each genetic code separately. The median was derived from $max(F_{\pi })$ values that were obtained from substitution models generating the given stationary nucleotide distribution. The lines are the best approximation based on generalized additive models with integrated smoothness estimation.

**Figure 8 ijms-24-01185-f008:**
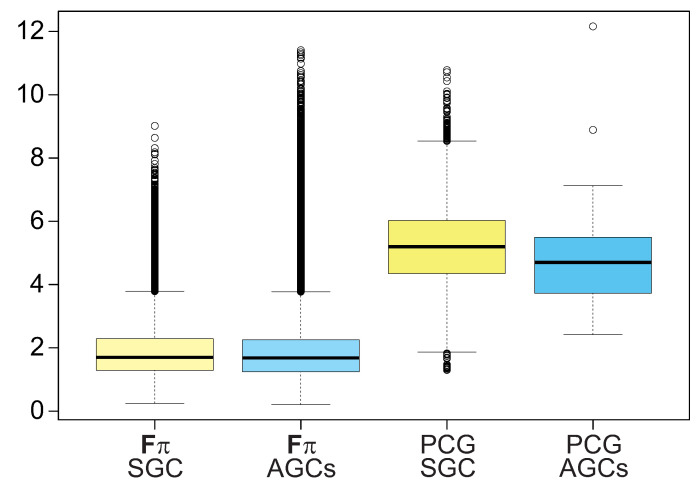
The deviation from the expectation in the codon usage for the 4FD codon groups calculated for protein-coding sequences translated by the standard genetic code (PCG SGC) and alternative genetic codes (PCG AGCs) in comparison to the selection strength on the 4FD codon usage calculated for the theoretical mutational pressures under the standard genetic code ($F_{\pi }$ SGC) and alternative genetic codes ($F_{\pi }$ ACGs). The thick line indicates median, the box shows quartile range and the whiskers denote the range without outliers.

**Table 1 ijms-24-01185-t001:** The descriptive statistics: the minimum ($min(F_{\pi })$), the median ($me(F_{\pi })$) and the maximum ($max(F_{\pi })$) of the selection strength on the 4FD codon usage ($F_{\pi }$) computed over 88,560 mutational pressures. Corresponding percent changes in relation to the SGC are also shown (%). The genetic codes were numbered according to the NCBI database (https://www.ncbi.nlm.nih.gov/Taxonomy/Utils/wprintgc.cgi), accessed on 10 November 2022. See Materials and Methods for details.

Code	$\min(F_{\pi })$	$\mathrm{me}(F_{\pi })$	$\max(F_{\pi })$	%$\min(F_{\pi })$	%$\mathrm{me}(F_{\pi })$	%$\max(F_{\pi })$
1 (SGC)	0.245	1.696	9.016	0.0	0.0	0.0
2	0.43	1.968	10.45	75.4	16.0	15.9
3	0.266	1.712	7.983	8.7	0.9	−11.5
4	0.24	1.627	7.855	−2.0	−4.1	−12.9
5	0.232	1.565	7.264	−5.4	−7.7	−19.4
6	0.271	1.734	8.926	10.5	2.2	−1.0
9	0.222	1.337	6.208	−9.3	−21.2	−31.1
10	0.249	1.537	8.341	1.6	−9.4	−7.5
12	0.317	1.91	9.556	29.3	12.6	6.0
13	0.33	1.759	7.671	34.7	3.7	−14.9
14	0.227	1.341	6.176	−7.4	−20.9	−31.5
16	0.297	1.674	8.344	21.2	−1.3	−7.5
21	0.203	1.371	6.758	−17.0	−19.2	−25.0
22	0.376	2.322	11.34	53.5	36.9	25.8
23	0.274	1.985	11.41	12.0	17.0	26.5
24	0.233	1.588	7.728	−5.0	−6.4	−14.3
25	0.243	1.726	9.002	−0.8	1.7	−0.2
26	0.406	1.894	9.737	65.7	11.7	8.0
27	0.262	1.73	9.052	7.0	2.0	0.4
28	0.262	1.73	9.052	7.0	2.0	0.4
29	0.262	1.537	8.242	7.0	−9.4	−8.6
30	0.238	1.708	8.831	−2.9	0.7	−2.1
31	0.256	1.708	7.742	4.6	0.7	−14.1
33	0.248	1.608	7.42	1.3	−5.2	−17.7

**Table 2 ijms-24-01185-t002:** The stationary distributions of four nucleotides generated by mutational pressures, which, together with the selection at the amino acid level, produce the minimum values of selection strength $F_{\pi }$.

Code	A	T	C	G	$\min(F_{\pi })$
1 (SGC)	0.05	0.07	0.2	0.68	0.245
2	0.05	0.12	0.66	0.17	0.430
3	0.05	0.05	0.21	0.69	0.266
4	0.05	0.06	0.17	0.72	0.240
5	0.05	0.06	0.7	0.19	0.232
6	0.05	0.06	0.17	0.72	0.271
9	0.05	0.07	0.31	0.57	0.222
10	0.05	0.09	0.29	0.57	0.249
12	0.05	0.07	0.71	0.17	0.317
13	0.05	0.07	0.71	0.17	0.330
14	0.05	0.05	0.21	0.69	0.227
16	0.06	0.05	0.23	0.66	0.297
21	0.05	0.06	0.68	0.21	0.203
22	0.06	0.07	0.41	0.46	0.376
23	0.05	0.08	0.29	0.58	0.274
24	0.06	0.09	0.41	0.44	0.233
25	0.05	0.06	0.2	0.69	0.243
26	0.07	0.13	0.32	0.48	0.406
27	0.05	0.09	0.21	0.65	0.262
28	0.05	0.09	0.21	0.65	0.262
29	0.05	0.13	0.26	0.56	0.262
30	0.05	0.05	0.15	0.75	0.238
31	0.05	0.05	0.83	0.07	0.256
33	0.05	0.05	0.16	0.74	0.248

**Table 3 ijms-24-01185-t003:** The stationary distributions of four nucleotides generated by mutational pressures, which, together with the selection at the amino acid level, produce the maximum values of selection strength $F_{\pi }$.

Code	A	T	C	G	$\max(F_{\pi })$
1 (SGC)	0.84	0.05	0.05	0.06	9.01596
2	0.81	0.07	0.06	0.06	10.445
3	0.15	0.73	0.06	0.06	7.98304
4	0.22	0.67	0.05	0.06	7.85467
5	0.17	0.71	0.05	0.07	7.26361
6	0.11	0.77	0.05	0.07	8.92571
9	0.11	0.77	0.05	0.07	6.2084
10	0.09	0.8	0.05	0.06	8.34086
12	0.07	0.81	0.06	0.06	9.55644
13	0.07	0.82	0.05	0.06	7.67111
14	0.22	0.67	0.05	0.06	6.17645
16	0.82	0.07	0.05	0.06	8.34363
21	0.12	0.77	0.05	0.06	6.75813
22	0.84	0.05	0.05	0.06	11.3414
23	0.84	0.05	0.05	0.06	11.4083
24	0.09	0.8	0.05	0.06	7.72787
25	0.78	0.11	0.05	0.06	9.0016
26	0.09	0.8	0.05	0.06	9.73709
27	0.21	0.68	0.05	0.06	9.05196
28	0.21	0.68	0.05	0.06	9.05196
29	0.19	0.69	0.06	0.06	8.24171
30	0.84	0.05	0.05	0.06	8.83112
31	0.1	0.79	0.05	0.06	7.74191
33	0.14	0.75	0.05	0.06	7.41951

**Table 4 ijms-24-01185-t004:** The genetic codes tested in the study. They are numbered according to the NCBI database (https://www.ncbi.nlm.nih.gov/Taxonomy/Utils/wprintgc.cgi), accessed on 10 November 2022, where you can find more detailed information about these coding systems.

Number	Code Description
1	Standard
2	Vertebrate Mitochondrial
3	Yeast Mitochondrial
4	Mold, Protozoan and Coelenterate Mitochondrial and *Mycoplasma*/*Spiroplasma*
5	Invertebrate Mitochondrial
6	Ciliate, Dasycladacean and Hexamita Nuclear
9	Echinoderm and Flatworm Mitochondrial
10	Euplotid Nuclear
12	Alternative Yeast Nuclear
13	Ascidian Mitochondrial
14	Alternative Flatworm Mitochondrial
16	Chlorophycean Mitochondrial
21	Trematode Mitochondrial
22	*Scenedesmus obliquus* Mitochondrial
23	*Thraustochytrium* Mitochondrial
24	Rhabdopleuridae Mitochondrial
25	Candidate Division SR1 and Gracilibacteria
26	*Pachysolen tannophilus* Nuclear
27	Karyorelict Nuclear
28	*Condylostoma* Nuclear
29	*Mesodinium* Nuclear
30	Peritrich Nuclear
31	*Blastocrithidia* Nuclear
33	Cephalodiscidae Mitochondrial UAA-Tyr

## Data Availability

The computations were conducted using C++ programming language. All source codes and raw data relevant to our investigations were included in Appendix A.

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
