# Peer review of "The Influence of the Selection at the Amino Acid Level on Synonymous Codon Usage from the Viewpoint of Alternative Genetic Codes"

_ijms, 2023, doi:10.3390/ijms24021185_

Round 1

Reviewer 1 Report

It is a nice paper, as usual, from the Wroclaw group. I have problems with results in 2.3. I just got lost on what you are looking at, and what is the result. What is on the Y axes in Figure 3? I think I do not understand that figure and its implication.

Can you make it clearer, like in 2.1 and 2.2. Those I can follow.

P2. „to use codons for which there more tRNA molecules in a cell” -> “to use codons for which there are more tRNA molecules in a cell”

P4. “rest 13 investigated” -> „rest of the 13 investigated”

P4 „the code 22 in green alga Scenedesmus obliquus, the code 23 in protist labyrinthulid Thraustochytrium and the code 2 in vertebrates, whereas the code 12 and 26 is an alternative to the yeast nuclear genome” -> „code 22 in the green alga Scenedesmus obliquus, code 23 in labyrinthulid protists, Thraustochytrium, and code 2 in vertebrates, whereas code 12 and 26 is an alternative to the yeast nuclear genome”

P4 When you speak about how TGA can mutate from Gly to something else, I think you wanted to write GGA (Gly) instead of GGG.

P5 „ is smaller for 9, 14 and 21 codes” -> „ is smaller for codes 9, 14 and 21”

P5 „ In turn, the 23 code has the TTA codon” -> „ In turn, code 23 has the TTA codon”

P5 please replace all -> with a → they look more professional and also do not fall into separate lines (see one of the last lines on page 5) Also in Figure 3

P6 „the 26 code” -> „code 26”

P6 “Due this substitution” -> “Due to this substitution”

P7 “with the code 21” -> “with code 21”

P7 “the code 23” -> “code 23”

P10 “a very low content both adenine and thymine” -> “a very low adenine and thymine content”

P12 “there are ten AGCs that generate the 4FD codon bias smaller than the SGC” -> “there are ten AGCs that generate 4FD codon biases smaller than the SGC”

P12 “clearly related with the mutational pressure” -> “clearly related to the mutational pressure”

P12 “it should be emphasized the diversified” -> “it should be emphasized that the diversified”

P12 “that in even free-living bacteria” -> “that even in free-living bacteria”

P13 “unrestrected” -> “unrestricted”

References:

Journal names should be capitalized like a title of a book, i.e. Journal of Molecular Evolution, Molecular Biology Reports, Nucleic Acids Research, etc.

Journal names should either be abbreviated or not abbreviated. Check IJMS style guideline.

Paper titles should not be capitalized like a journal name.

Ref 2’s DOI is 10.1038/nrg2899

Ref 11: If you otherwise do not abbreviate journal names, then do not do it here either. Also write out the full range of pages, i.e. 409–416. DOI is 10.1101/gr.9.5.409

Ref 15. DOI 10.1016/S0021-9258(19)81068-2. Full range of pages!

Ref 16 missing doi

Ref 17 missing doi

Ref 22. Article identifier is e1001004

Ref 34. No doi and should have full page range

From ref 34 there is no doi given till ref 67

Ref 38,40,54,55,67,83. Full range of pages

Ref 45,73,76,80 missing article identifier

Ref 47 delete text from the end about citation count

Author Response

Reviewer 1

Reviewer’s remark

It is a nice paper, as usual, from the Wroclaw group.

Authors’ answer

Thank you very much for your comments and suggestions. We included all of them, which substantially improved our manuscript.

Reviewer’s remark

I have problems with results in 2.3. I just got lost on what you are looking at, and what is the result. What is on the Y axes in Figure 3? I think I do not understand that figure and its implication. Can you make it clearer, like in 2.1 and 2.2. Those I can follow.

Authors’ answer

The aim of the chapter 2.3 is to show how the mutational pressure influences the effect of the selection strength on the codon usage. We changed the title of this chapter and clarified the description. The Y axes in this figure represent individual nucleotide substitution rates. We explained it in the legend.

Reviewer’s remark

P2. „to use codons for which there more tRNA molecules in a cell” -> “to use codons for which there are more tRNA molecules in a cell”

P4. “rest 13 investigated” -> „rest of the 13 investigated”

P4 „the code 22 in green alga Scenedesmus obliquus, the code 23 in protist labyrinthulid Thraustochytrium and the code 2 in vertebrates, whereas the code 12 and 26 is an alternative to the yeast nuclear genome” -> „code 22 in the green alga Scenedesmus obliquus, code 23 in labyrinthulid protists, Thraustochytrium, and code 2 in vertebrates, whereas code 12 and 26 is an alternative to the yeast nuclear genome”

P5 „ is smaller for 9, 14 and 21 codes” -> „ is smaller for codes 9, 14 and 21”

P5 „ In turn, the 23 code has the TTA codon” -> „ In turn, code 23 has the TTA codon”

P5 please replace all -> with a → they look more professional and also do not fall into separate lines (see one of the last lines on page 5) Also in Figure 3

P6 „the 26 code” -> „code 26”

P6 “Due this substitution” -> “Due to this substitution”

P7 “with the code 21” -> “with code 21”

P7 “the code 23” -> “code 23”

P10 “a very low content both adenine and thymine” -> “a very low adenine and thymine content”

P12 “there are ten AGCs that generate the 4FD codon bias smaller than the SGC” -> “there are ten AGCs that generate 4FD codon biases smaller than the SGC”

P12 “clearly related with the mutational pressure” -> “clearly related to the mutational pressure”

P12 “it should be emphasized the diversified” -> “it should be emphasized that the diversified”

P12 “that in even free-living bacteria” -> “that even in free-living bacteria”

P13 “unrestrected” -> “unrestricted”

Authors’ answer

We included all your corrections in the manuscript.

Reviewer’s remark

P4 When you speak about how TGA can mutate from Gly to something else, I think you wanted to write GGA (Gly) instead of GGG.

Authors’ answer

You are wright. Thank you for this insightful comment. We corrected it.

Reviewer’s remark

Journal names should be capitalized like a title of a book, i.e. Journal of Molecular Evolution, Molecular Biology Reports, Nucleic Acids Research, etc.

Journal names should either be abbreviated or not abbreviated. Check IJMS style guideline.

Paper titles should not be capitalized like a journal name.

Ref 2’s DOI is 10.1038/nrg2899

Ref 11: If you otherwise do not abbreviate journal names, then do not do it here either. Also write out the full range of pages, i.e. 409–416. DOI is 10.1101/gr.9.5.409

Ref 15. DOI 10.1016/S0021-9258(19)81068-2. Full range of pages!

Ref 16 missing doi

Ref 17 missing doi

Ref 22. Article identifier is e1001004

Ref 34. No doi and should have full page range

From ref 34 there is no doi given till ref 67

Ref 38,40,54,55,67,83. Full range of pages

Ref 45,73,76,80 missing article identifier

Ref 47 delete text from the end about citation count

Authors’ answer

We corrected the references accordingly. Thank you for your detailed correction.

Reviewer 2 Report

The current manuscript entitled " The influence of the selection at amino acid level on synonymous

codon usage from viewpoint of alternative genetic codes" provides a new way to look at the synonymous codon bias resulted from selection strength on SGC and AGCs, which can be of interest to the evolutionary and genetic communities. However, there are major and minor comments that I believe needs to be addressed in the manuscript to be qualified for publication.

Major comments:

1.       I wonder whether the 23 AGCs tested in the study (Table 4) are representative? The author should explain.

Minor comments:

1.       The detailed definition of Fπ should be given not only in the M&M but also in the first place in this manuscript.

2.       Should the author give more visualization of Table 1 to explain results?

3.       In the Figure3 legend, significance tests should be performed.

4.       All source codes and raw data should be provided in supplementary material

Author Response

Reviewer 2

Reviewer’s remark

The current manuscript entitled " The influence of the selection at amino acid level on synonymous

codon usage from viewpoint of alternative genetic codes" provides a new way to look at the synonymous codon bias resulted from selection strength on SGC and AGCs, which can be of interest to the evolutionary and genetic communities. However, there are major and minor comments that I believe needs to be addressed in the manuscript to be qualified for publication.

Authors’ answer

Thank you very much for your comments and suggestions. We included all of them, which improved our manuscript.

Reviewer’s remark

Major comments:

1. I wonder whether the 23 AGCs tested in the study (Table 4) are representative? The author should explain.

Authors’ answer

We studied all alternative genetic codes available at the NCBI site that encode 20 canonical amino acids and differ from the SGC in at least one assignment of amino acids and/or stop to codons. At the same time, these codes contain the same 4FD codons for Ala, Gly, Pro, Thr and Val as the SGC. We clarified it in the manuscript in Introduction as well as Material and methods. We also commented their representativeness. New codes are still discovered especially in poorly studied groups of protists. Therefore, we cannot exclude that other codes can show a more extreme influence on the codon usage.

Reviewer’s remark

Minor comments:

1. The detailed definition of Fπ should be given not only in the M&M but also in the first place in this manuscript.

Authors’ answer

At the beginning of Results, we described how this parameter was calculated.

Reviewer’s remark

2. Should the author give more visualization of Table 1 to explain results?

Authors’ answer

Following your suggestion, we added a figure showing the ranking of alternative genetic codes according to percent changes in the median selection strength on the 4FD codon usage in the relation to the SGC.

Reviewer’s remark

3. In the Figure3 legend, significance tests should be performed.

Authors’ answer

Following your remark, we added the results of statistical tests.

Reviewer’s remark

4. All source codes and raw data should be provided in supplementary material

Authors’ answer

We included all source codes and raw data in the supplementary material.

Reviewer 3 Report

Pawlak, et al

IJMS- “The influence of the selection at amino acid level..”

The point of the extensive analysis in this ms is: mutation of codons yields different amino acids. Therefore, the properties of the mutated amino acids should be considered in reasoning about the probability of an accepted codon mutation.

The problem is that it is not clear when, or even if, the result of a mutation is significant enough to feed back to evolutionarily-accepted codon usage. The same authors have considered this problem before (Błażej et al. 2017), and have discussed the magnitude of such feedback. Explicit discussion of the relative significance of the effects in this new work, which adds alternative codes to the analysis, is needed to enable a reader to evaluate the evolutionary significance of new findings here.

Similarly, the differences between alternative and standard genetic codes detected here might result from differences in the pressures on genomes in different environments. It would be extremely helpful to have a specific discussion of what the detected differences ultimately mean about translation systems.

Semi-finally, the Discussion attempts to interpret present research findings in terms of alternative models for message evolution: e.g., mutation pressure vs amino acid stereochemistry acting on substitution. In its submitted form, this discussion (presumably the major conclusion of the entire work) is so vague that, despite several, readings, I cannot say what the authors think are their final results. This critical text needs to be greatly clarified.

The paper urgently needs proofing for errors of grammar and spelling, which make it notably hard to understand. For example, on page 2, 4 it is deduced that GGA should be a frequent Gly codon, whereas it seems to me from the text that GGA should be a rare one. Self-contradictory text will inhibit a naïve reader (or a sophisticated one), to say the least.

Błażej P, Mackiewicz D, Wnętrzak M, Mackiewicz P. 2017. The Impact of Selection at the Amino Acid Level on the Usage of Synonymous Codons. G3 Bethesda Md 7: 967–981.

Author Response

Reviewer 3

Reviewer’s remark

The point of the extensive analysis in this ms is: mutation of codons yields different amino acids. Therefore, the properties of the mutated amino acids should be considered in reasoning about the probability of an accepted codon mutation.

The problem is that it is not clear when, or even if, the result of a mutation is significant enough to feed back to evolutionarily-accepted codon usage. The same authors have considered this problem before (Błażej et al. 2017), and have discussed the magnitude of such feedback. Explicit discussion of the relative significance of the effects in this new work, which adds alternative codes to the analysis, is needed to enable a reader to evaluate the evolutionary significance of new findings here.

Authors’ answer

Thank you very much for your comments and suggestions. We included all of them, which significantly improved our manuscript. Following your remark, we calculated the strength of selection at the amino acid level on the 4FD codons in real protein-coding sequences that are translated by the studied genetic codes. We added a new chapter presenting these results. The analysis showed that this effect is of the same order of magnitude as the theoretical values and cannot be neglected. We also more widely discussed the significance of this effect.

Reviewer’s remark

Similarly, the differences between alternative and standard genetic codes detected here might result from differences in the pressures on genomes in different environments. It would be extremely helpful to have a specific discussion of what the detected differences ultimately mean about translation systems.

Authors’ answer

We discussed the importance of the studied effect on the protein translation systems, too.

Reviewer’s remark

Semi-finally, the Discussion attempts to interpret present research findings in terms of alternative models for message evolution: e.g., mutation pressure vs amino acid stereochemistry acting on substitution. In its submitted form, this discussion (presumably the major conclusion of the entire work) is so vague that, despite several, readings, I cannot say what the authors think are their final results. This critical text needs to be greatly clarified.

Authors’ answer

We modified and extended the Discussion section to clarify our findings and implications of the studied effect of selection of the synonymous codon usage. We hope that the descriptions are much clearer.

Reviewer’s remark

The paper urgently needs proofing for errors of grammar and spelling, which make it notably hard to understand. For example, on page 2, ¶4 it is deduced that GGA should be a frequent Gly codon, whereas it seems to me from the text that GGA should be a rare one. Self-contradictory text will inhibit a naïve reader (or a sophisticated one), to say the least.

Authors’ answer

We corrected the manuscript in terms of grammar and spelling including the description mentioned by the Reviewer about the frequency of glycine codons. We hope that the description is consistent with other part of the manuscript.